



# Variations in land types detected using methane retrieved from space-borne sensor

Saheba Bhatnagar[1]*, Mahesh Kumar Sha[2]**, Laurence Gill[1], Bavo Langerock[2], Bidisha Ghosh[1]

[1]School of Engineering, Trinity College, Dublin, Ireland
[2]Royal Belgian Institute for Space Aeronomy (BIRA-IASB), Brussels, Belgium

*Correspondence to*:  Saheba Bhatnagar (sbhatnag@tcd.ie) Mahesh K Sha (mahesh.sha@aeronomie.be)

**Abstract.** Methane ($CH_4$), a potent greenhouse gas, traps heat in the atmosphere and significantly contributes to global warming. Atmospheric $CH_4$ comes from various natural and anthropogenic sources. $CH_4$ emissions from the decomposition of organic material by bacteria in natural wetlands, other land types, agriculture, and waste management constitute the major component of global emissions. Although there is no clear evidence that $CH_4$ emissions from wetlands and other natural sources have increased substantially in the last decade, uncertainties remain regarding sources and their spatial extent causing discrepancies between emission estimates from inventories/models and estimates inferred by an ensemble of atmospheric inversions. Here we show that satellite-based $CH_4$ total column measurements along with surface albedo from Sentinel-5 Precursor (S-5p) show unique sensitivity to certain land types. Consequently, the areal extent of six land types (marsh, swamp, forest, grassland, cropland, and barren-land) could be identified with high overall accuracy by analysing S-5p data over Canada utilising our classification-segmentation algorithm. Monthly and yearly inventory maps were created, which can be used to validate or complement global models where data from other sources are missing and may help in further constraining the methane budget.

## 1 Introduction

Methane ($CH_4$), after carbon dioxide ($CO_2$), is the second most important anthropogenic greenhouse gas contributing to climate change. Compared to $CO_2$, it has a shorter atmospheric lifetime of about 9 years (Prather et al., 2012), making it a favourable target for climate change mitigation. Atmospheric emissions and concentrations of $CH_4$ have increased continuously over the last decade (Saunois et al., 2020). Wetlands are known to be the largest natural source of $CH_4$, with an estimated average global emission, from "bottom-up" inventories/modelling approaches, of 149 Tg $CH_4$ yr$^{-1}$ (range 102-182) during the past decade (2008-2017) (Saunois et al., 2020). This represents about 20% of the total $CH_4$ emission sources estimated by such approaches. The wide variability in estimates results from the difficulty in defining $CH_4$ producing wetland areas and parameterising terrestrial anaerobic conditions that drive $CH_4$ sources and conversely, oxidative conditions leading to $CH_4$ sinks (Melton et al., 2013; Poulter et al., 2017; Wania et al., 2013). However, average $CH_4$ bottom-up wetlands' emission estimates are lower than top-down emission estimates of 181 Tg $CH_4$ yr$^{-1}$ (range 159-200) inferred by an ensemble of atmospheric inversions using an atmospheric constraint (Saunois et al., 2020). The difference between average wetland emissions from bottom-up and top-



down estimates has increased from about 17 to 30 Tg CH$_4$ yr$^{-1}$ in the recent global methane budget study (Saunois et al., 2016; Saunois et al., 2020). This difference for other natural emission sources (e.g., inland waters, geological, permafrost, vegetation, etc.) is 185 Tg CH$_4$ yr$^{-1}$ (Saunois et al., 2020). Reducing the differences between the two estimate methods is of prime importance to constrain the global methane budget more accurately. About 5% of the atmospheric CH$_4$ uptake is by the methanotrophic bacteria present in unsaturated oxic soil, with the main sink being chemical reactions in the atmosphere (Saunois et al., 2020). The CH$_4$ emission contribution from land types is calculated as the product of emission flux density and the surface extent of CH$_4$ source/sink area (Bohn et al., 2015; Melton et al., 2013). The seasonal and inter-annual variability of these land types' areal extent is considered the main cause of uncertainty in calculating their absolute flux of CH$_4$ emissions, which is significant for the global CH$_4$ budget (Bohn et al., 2015; Desai et al., 2015; Poulter et al., 2017). However, equally the actual areal extent of different wetland types is still very approximate and in many parts of the world and needs to be improved, such as the areas of marsh, bog, swamp and fen in Canada (Comer et al., 2000; Harris et al., 2021). Here, we show for the first time that a space instrument can detect the sensitivity of CH$_4$ product to variations in land types, i.e., TROPOspheric Monitoring Instrument (TROPOMI), which is onboard the European Space Agency's (ESA) Sentinel-5 Precursor (S-5p) satellite measuring daily global total column concentrations of atmospheric CH$_4$. We further show how this information can be used to identify the inherent sensitivities amongst land types responsible for such positive or negative emissions, but also help to better define the areal extent of the different land use types (particularly wetlands) from which more accurate greenhouse gas global budgets can be calculated

## 2 Data and study region

### 2.1 Satellite-based CH$_4$ total column data used in this study

S-5p is orbiting the Earth in a near-polar sun-synchronous orbit at an altitude of 824 km with an ascending node equator crossing at 13:30 local time since its launch on 13 October 2017. TROPOMI is a nadir-viewing grating spectrometer measuring the solar radiation reflected by the Earth and its atmosphere in eight spectral bands from the ultraviolet (UV) to the short-wave infrared range (SWIR). S-5p has an orbit cycle of 16 days and covers the Earth with 14 orbits per day. The push-broom configuration with the imaging capabilities allows a wide swath of 2,600 km, which results in daily global coverage. The vertically integrated abundances of CH$_4$ are retrieved from the SWIR (2305-2385 nm) spectral channel (Veefkind et al., 2012). The CH$_4$ total column measured by the satellite is a combination of CH$_4$ production, oxidation in the atmosphere (or soil uptake), and transport. The spatial resolution of the operational level 2 SWIR product was originally 7×7 km$^2$ in exact nadir and was increased to 5.5×7 km$^2$ on 6 August 2019. The operational processing to retrieve the column averaged dry air mixing ratio of CH$_4$ is performed by RemoTeC S5 algorithm (Hu et al., 2016). The operational CH$_4$ total column product consists of a standard product and a bias-corrected product. The details of the bias correction are described in the Algorithm Theoretical Baseline Document (ATBD) (Hu et al., 2016). The latest product version of the S-5p CH$_4$ total column data from Jan 2018 until Dec 2019 has been used in this study. This period also includes the data during the satellite's commissioning phase (Jan



– end April 2018). The quality of the data has been verified by the ESA mission performance centre (MPC) by performing
validation against reference ground-based remote sensing networks of the Total Carbon Column Observing Network (TCCON)
and the Infrared Working Group (IRWG) of the Network for the Detection of Atmospheric Composition Change (NDACC)
(Langerock & Sha, 2019; Sha et al., 2021). The reported systematic uncertainty of the bias corrected methane product validated
against 25 TCCON stations is -0.26±0.56 % and the random uncertainty is 0.57±0.18 %. These stations are located in different
parts of the world representing different surface conditions (land types and corresponding surface albedos) and atmospheric
conditions, As S-5p records solar absorption measurements reflected by the Earth's surface and the atmosphere, measurements
are not possible over larger parts of Canada during the winter months (Nov-Jan).

The S-5p bias-corrected $CH_4$ total column values along with the retrieved surface albedo (SA) for quality assurance (qa) value
greater than 0.5 were selected and binned on a regular 0.05° grid to form the level 3 (L3) data. The harp component of the
ESA atmospheric toolbox (https://atmospherictoolbox.org) was used to perform the latitude longitude regridding where each
S-5p pixel contributes to the regridded $CH_4$ value of the target grid cell if there was an overlap of the pixel and the grid cell.
In case when multiple pixels overlap, a grid cell weighted average was taken using the overlap area as the weight.

## 2.2 Region of study

### 2.2.1 Selection of the region

Wetlands cover approximately 5.5% of the global land surface with an average areal extent of 8.0 to 8.4 million $km^2$. Apart
from the ecological significance, wetlands store atmospheric carbon and act as a carbon sink. Peatland wetlands, for example,
cover 3% of the Earth's land surface but store approximately 25% of the global soil carbon (Yu et al., 2011). The $CH_4$
production in wetlands is influenced by the spatial and temporal extent of anoxia (water level in the soil), temperature,
availability of substrate, and plant ecology (Valentine et al., 1994; Wania et al., 2010; Whalen, 2005; Swenson et al., 2019).
Monitoring these wetlands using remote sensing is a resource and time-efficient endeavour with significant ecological and
environmental importance. A large section (~25%) of the world's remaining wetlands are located in Canada, covering 12.9%
of Canada's terrestrial area (National Wetlands Working Group, 1997; Environmental and Climate Change Canada, 2016).
Therefore, we have chosen Canada as our study region due to the presence of large wetland areas (and other land types) which
are known to emit differing quantities of $CH_4$ and the availability of a land type map for Canada (described in section 2.2.2)
for verifying our results.

### 2.2.2 Canadian Wetland Inventory

In 2019, Amani et al. (2019) created the first Canada wetland inventory (CWI) using a composite of approximately 30,000
Landsat-8 surface reflectance images collected from 2016 to 2018. This method allows monitoring and mapping wetlands
every three years with 66% producer and 63% user accuracy. The CWI map included five wetland classes defined by the
Canadian Wetland Classification System (CWCS) – bog, fen, marsh, swamp, and shallow-water – as well as other land types





– forest, grassland, cropland, barren (rocks, gravel, built-up areas, non-vegetation), deep-water, and snow (Fig. 1a left inset). The S-5p data being used for this study matches the timeframe of creation of the first CWI; therefore, it was used to generate the S-5p resolution specific ground truth labels described in the next section.

## 3 Methods

A brief description of the machine learning (ML) algorithm utilised to create the labels and analyse the satellite data is
described here. The ML algorithm used was initially developed to identify vegetation communities within wetlands using remote sensing, and the steps for customising the algorithm for detecting the sensitivities of land types to the S-5p products (methane and SA) are given here.

### 3.1 Creating ground truth labels from CWI

The CWI map is available at a significantly higher spatial pixel resolution of 30 m compared to the binned S-5p resolution at
0.05° grid (~5.5 km). The CWI map was therefore upscaled to a lower resolution map combining additional Moderate Resolution Imaging Spectroradiometer (MODIS) normalised difference vegetation index (NDVI) product. The MODIS NDVI (Normalized Difference Vegetation Index) (MOD13A3) produces monthly NDVI maps at 1 km resolution with about 15 tiles covering the area of interest (AOI) in Canada marked with a red rectangular box in Fig. 1. All of the NDVI images were mosaicked using the mean value (for the overlapping areas) for the 24 months of the study period. Therefore, a 3-dimensional
image with 24 bands was created using layer stacking for the AOI. To create proper segments, the CWI map was also upscaled to 1 km resolution such that it is compatible with the MODIS NDVI product. The map was upscaled to 5.5 km spatial resolution using nearest neighbour interpolation for upscaling it to the same resolution as the L3 S-5p data.

Some of the islands (far north) were not considered due to poor or insufficient availability of S-5p data during long winter periods, in which the area was covered in snow/ice and/or clouds, limiting the satellite's view. Due to the current unavailability
of methane data over water from TROPOMI, land types like shallow water and deep water were also not considered for this study. The smoothed segmented map created for the selected area (Fig. 1a right inset) was used as the ground truth (GT) in this paper. The conventional remote sensing analysis is performed against manually collected field data. Other ready to use products like CORINE land cover (CLC 2018), MODIS land cover map (MCD12Q2: https://lpdaac.usgs.gov/products/mcd12q2v006/), etc., can also be used as initial ground truth for validating a similar study. It
has to be noted that in this study, the upscaling was due to the absence of field-based ground truth. The MODIS land cover map also had a limitation on the wetlands types; therefore, CWI was the best product available to conduct this study.

### 3.2 Classification-segmentation machine learning algorithm

The level 3 regridded S-5p $CH_4$ total column and SA over Canada were analysed utilising a ML algorithm described in Bhatnagar et al. (2020). The dendrogram created using the CWI describes the degree of dissimilarity in $CH_4$ between the



clusters of land types (Fig. 1b). This dissimilarity was measured in the form of Euclidean distance between the centroids – depicting how close/far (in terms of $CH_4$ total column values) the land types exist. The workflow, including the development of ground truth (GT) maps, creation of monthly and yearly maps, and performance evaluation of the algorithm, is described in Fig. 2.

The first step of the analysis was creating a GT map for evaluating the sensitivity of S-5p $CH_4$ total column measurements to
certain land types, especially wetlands. Next, the S-5p data was used for classification using a segmentation model using random forest classification followed by graph cut segmentation based on posterior probability; a detailed description of the model can be found in Bhatnagar et al. (2020). Therefore, both pixel-based intensity and contextual information (area-based segmentation) were utilised. For training, 30% of stratified random samples (pixels) from the GT map were used. Collins et al. (2020) suggest that using random training samples with equal representation of each class is necessary to avoid classifiers'
bias. The manual selection of training points may produce clustered training points, thereby increasing the inherent spatial autocorrelation (Millard and Richardson, 2015). Therefore, stratified random sampling of training data with equal weightage to each class was selected for this study. Stratified random sampling is advantageous as it usually yields more accurate estimations (Stumpf et al., 2013).

Using the segmentation model mentioned above, every pixel under AOI was mapped at least once every month. Therefore, a
total of 679 daily maps were created for the years 2018 and 2019. It has to be noted that this study does not use conventional time-series analysis. Here, every image was treated individually with equal realisation, i.e., all the images under consideration had equal importance. The majority voting was done for each pixel in the daily maps to create the monthly maps using Eq.1 (Jimenez et al. (1999)), i.e., for every pixel $p \,\epsilon\, N$ a class $x \,\epsilon\, n$ would be assigned if,

$$\sum_{p=1}^{N} \widehat{F_p}(x) \;=\; max_{x=1}^{n} \sum_{p=1}^{N} F_p(x) \tag{1}$$

where $N$ are the total number of pixels, and $\widehat{F_p}(x)$ is the majority voted map at the end of each month for the years 2018, 2019.
Pixels that were not mapped for any given day for that month were removed. Furthermore, only the covered/mapped area was used for further accuracy analysis for each land type (class). Class Accuracy (CA) is the ratio of the diagonal vector of the class under consideration with the total number of pixels belonging to the same class, shown in Eq.2.

$$CA = \frac{TP}{TP + FN} \tag{2}$$

Eventually, a majority voting was applied on monthly maps (separately for two years) to obtain the final annual aggregated
map. This map gives an idea of the difference in land types and opens an area of application of S-5p products for this purpose. For every month, only the pixels with CA≥55% were selected to form a high-confidence map for that month and classes like bog, fen, deep water, shallow water were omitted in these maps due to low CA values. The monthly high-confidence maps were again combined (using majority voting) to form the final aggregated map for a year. The performance of the algorithms





was tested on the remaining pixels. A kappa value of 0.69 was achieved, which are comparable to the kappa value (0.66)
reported by Amani et al. (2019).

The accuracy of areal detection should not only be checked by typical PB evaluation metrics such as producer and user accuracy
(Story & Congalton, 1986). For change detection, PB comparison may not provide the true extent of error; therefore, segment-
based comparison based on the area's geometry was needed (Bhatnagar et al., 2021). Hence, a set of error metrics linked with
location and extent of land type detection was calculated for the annual maps for both years while comparing it with the GT
of the captured region (Bhatnagar et al., 2021).

### 3.3 Error metric

The set of error metrics specific to measuring spatial changes to identify differences between $CH_4$ identification, used in this
study, is described below.

- Jaccard Similarity Index (J) measures the similarity between the members of the two sets and reports the amount of
similarity and distinction (Real & Vargas, 1996).

- Area (A) estimates the total area of the selected land type. The area of every individual pixel is determined by looking at
  its 2×2 neighbourhood. Each pixel is part of four different 2×2 neighbourhoods, which indicates the change in the overall
  growth/shrinkage of the community.

- Orientation (O) gives the angle between the x-axis and the ellipse's major axis (covering the entire land type). It can range
from -90 to + 90 degrees, indicating the direction of the land type change.

- Extent (E) indicates the ratio of total pixels present in the bounding box to the total pixels present in the image. The
  bounding box represents a box (rectangle/square) covering the major cluster of pixels present for a land type in an image.

## 4 Results and discussion

### 4.1 Seasonal and spatial variations of S-5p CH4 total column

Time-series of S-5p $CH_4$ total column concentrations over the four land types (marsh, swamp, forest, and grassland) are shown
in Fig. 1c, with gaps indicating missing data during November-January (winter months). Although the absolute total column
values indicate a lack of uniqueness, the entire dataset over two years reveals significant distinction, as seen in the dendrogram
plot of hierarchical clusters present within the dataset (Fig. 1b). Within the two main clusters of the dendrogram, major
wetlands and forest areas are segregated from other land types. The height of each dendrogram leg signifies the uniqueness of
S-5p $CH_4$ total column values from each land type. Fen appears closely related to marsh (based on the mean value (~300 km)
of the dendrogram leg). Swamp and forest also show inter-mixed features, most likely due to their spatial proximity. For similar
reasons, grass and barren-land types show indistinguishable features indicating that $CH_4$ total column values from
neighbouring land types may not be identifiable as distinct.



S-5p CH₄ total column and SA values were available for all key land types as classified in CWI map (Amani et al., 2019).

Area covered by snow, as obtained using MODIS snow product (Hall et al., 2006), interfered with the capture and visibility of land types decreased in all cases with increased snow cover (Fig. 3).

### 4.2 Detection of sensitivity of CH₄ to land types

The S-5p CH₄ total column and SA from gridded pixels were analysed together and separately using a classification-segmentation algorithm for each available day. Algorithm steps, training data (30%) and all details were described previously

in Section 3. The analysis generated daily maps from the testing data (70%) showing the extent of 8 different land types over 365 days in 2019 and 314 days in 2018 (51 missing days, mostly during the S-5p commissioning phase). The daily maps during a calendar month were combined to create a monthly map where each pixel was identified as the land type using majority voting as described in the previous section. The time-series of the class accuracy (CA) values for each land type calculated compared to the GT maps are shown in Fig. 3, along with the SA for the respective land types.

The performance of the algorithm for detecting area of each land type showed the sensitivity of S-5p CH₄ total column and SA to the land type in 2018 and 2019 as presented in Table 1, and the confusion matrix in Table 2. Table 1 depicts the six major land types. Since the overall CA of bog and fen were less than 55%; these land types were not considered in the final creation of annual maps (Fig. 4). Table 2 (confusion matrix) gives pixels for all eight land types for better understanding. The CA improved with the inclusion of SA data for most land types, especially for swamp and cropland compared to the analysis

considering only S-5p CH₄ data (Fig. 3 all sub-plots).

Using S5-p products, the land types with large areal extent, such as marsh, forest, grassland, swamp, and cropland, showed high detectability (CA>60%), while wetland types such as bog and fen showed low CAs due to low areal extent and proximity to other dominant land types. Bog, fen, and swamp were often misclassified due to their intermixed land distribution. In the winter months, CA decreased due to the lack of S-5p data and were omitted from the plots. Marsh was detected with the highest

CA, with variations in accuracy linked with a lack of available pixels. Similarly, for grassland and barren-land, CAs were linked with the area covered by S-5p, with grassland showing better detectability. The other key wetland-type, swamp, showed better detectability in spring and autumn than summer when it was misclassified as forest during the growing season. Forest was detected with reasonable OA, which slightly reduced with the melting of snow cover. The inclusion of SA improved the detectability of cropland significantly. The detectability of most dominant land types utilising CH₄ data thus indicates a

significant difference and sensitivities of CH₄ emissions between land types.

The seasonal variations of the CH₄ data and SA from land types are illustrated by delineating different land types during different seasons of 2018 (Fig. 4a) and 2019 (Fig. 4c). Land types detected with all CA values were plotted in monthly maps, with any missing or non-detectable pixels shown as white in Fig. 4. The sensitivities of the method for distinguishing between marsh, swamp, grassland, and barren-land were strongest during March-May, while sensitivities to cropland and neighbouring

land types were strongest during May-June. It has to be noted that the random uncertainty in the XCH₄ data, as mentioned earlier, is 10.5 ppb. When the difference between the CH₄ values is less than the uncertainty, the results are considered to be





erroneous. This can be seen for major land types in Figure 1c. The months March-May have a difference greater than 10.5 ppb which explains better detection of classes during that period.

The yearly maps for 2018, 2019 were created using majority voting of the monthly maps during the calendar year, only
including land types with CA>55% (Fig. 4b&d). The CA was calculated considering the classified map provided by Amani et al. (2019) and not validated against independent field assessment (due to unavailability of this information). The areal extent of marsh, swamp, forest, and grassland was identified with high confidence (Figure 3). The land types with large areal extent were generally detected well with high accuracy except in the case of barren-land, which showed low producer accuracy indicating its low sensitivity to $CH_4$ emissions. The bog and fen wetlands with lower areal extent were misclassified as marsh
(Table 2). Similarly, some pixels in the swamp were misclassified as forest during summer periods (June-August). This is mainly due to the proximity of the land types, which leads to pixel-mixing effects (also seen in Figure 1 (b)). Land types other than wetland (cropland and barren-land) were best identified in summer with good boundary delineation, and grassland, although adjacent to marsh, was well distinguishable throughout the year, indicating the sensitivity to the difference $CH_4$ for these land types.

It is possible that SA at 2.3 µm is particularly sensitive to some land types, leading to enhancement of identification (additional tests done using just SA information are shown in Fig. 5). Also, as seen in Fig. 3, the land types show very small changes in the SA retrieved in the 2.3 µm over time during the year.

In Fig. 5, the green line shows the SA case, plotted along with the bias-corrected XCH4 case (orange line) and the bias-corrected XCH4 + SA case (blue line) which are shown in the paper. Independently, $XCH_4$ or SA provides reasonable accuracy
for land type sensitivity detection. In particular, for land types where methane activities are expected to be fluctuating less, we tend to see a significant impact of SA on accuracy. When we use $CH_4$ and SA combined, we see the best performance in identifying the sensitivities of land types as in the proposed algorithm. Therefore, we find that the best choice is the combined usage of the bias-corrected XCH4 + SA case. In addition, the effect of the bias-correction applied to the XCH4 data was seen to be similar to standard XCH4 (std-XCH4) data for most land types see Fig. 6, where the dashed black line shows the standard
XCH4 case, plotted along with the bias-corrected XCH4 case (orange line) and the bias-corrected XCH4 + SA case (blue line). The largest difference in accuracies between the standard and the bias-corrected XCH4 runs can be seen for cropland. This is because the maximum change in SA was observed for cropland (Fig. 3). The other land types show very small changes in the SA retrieved in the 2.3 micron over time during the year. Therefore, the effect of the bias-correction applied to the XCH4 data is similar for those land types with similar SA conditions.

Detection of the areal extent of land types and the difference in their sensitivities based on S-5p $CH_4$ total column combined with SA was carried out utilising the proposed machine learning (ML) algorithm, where the efficiency of detection was investigated using a set of areal error metrics. Jaccard similarity index (J), area (A), orientation (O) and extent (E) (Table 3). Good detection was seen for all six key land types, while the variability of these metrics mainly was attributed to the lack of availability of S-5p pixels, which were often due to inimical meteorological conditions.





Lastly, this study aims to highlight the potential of S-5p products for detecting the variation in land types. Due to the absence of in situ measurements, the study could not verify the correlation of sensitivities of the land types as seen from space and actual ground coverage. The other limitation was the lack of field measurements in terms of land cover. This study is highly dependent on prior knowledge about the ground truth/land type locations. Furthermore, at least a coarse information of land type location is essential to conclude the sensitivity variations present. Whilst this methodology presented does not actually

quantify the methane fluxes from the different land types, with ever more focus and field studies globally on greenhouse gas emissions from different land types providing ground truth data, as well as advancing knowledge about atmospheric physics, it may soon be possible to fractionate the remotely-sensed net $CH_4$ signal via further modelling to be able to start to differentiate $CH_4$ emissions between the land types with more confidence using this product. Furthermore, having an additional remote sensing approach to map the areal extent of different land uses (which actually incorporates one of the main greenhouse gases)

helps to reduce the uncertainties in this areal parameter, which is used, in conjunction with typical greenhouse gas emissions for different land types from field data, to provide better estimates of the overall impact on global carbon budgets that these large, more remote areas of wetlands in the world, are having.

**Conclusions**

This work demonstrates that the S-5p $CH_4$ total column data with a machine learning algorithm can reveal unique sensitivity to certain land types, especially marsh, forest and grassland. Analysing such $CH_4$ data along with derived surface albedo, the areal extents of six land types (following CWI), including two major wetland types (marsh and swamp) covering ~60% of the total wetland area of Canada, were identified for two consecutive years 2018, 2019. As the vegetation appearance of land types, especially wetlands, can vary seasonally, mapping it solely using aerial photography or satellite imagery may lead to

errors because of a lack of consistent vegetation patterns (Environmental and Climate Change Canada (2016)). The CWI generated using S-5p data in this study is complementary to the traditional methods of land type identification showing daily, monthly, seasonal, and yearly changes. These maps can be used by the WAD2M (Wetland Area Dynamics for Methane Modeling) to either verify or complement their data where measurements from other sources are not available. The study presents an entirely new application of satellite-based $CH_4$ data illustrating its potential for land type identification of large

areas, monitoring and studying the dynamic change over time, and helping to constrain global methane emission models.

**Code availability**

The classification (Bhatnagar et al. 2020) code used for the study can be accessed via the GitHub repository https://github.com/saheba92/Mapping-vegetation-communities



**Data availability**

The publicly available S-5p CH₄ total column and SA data is available via https://s5phub.copernicus.eu/dhus/#/home, the S-5p data during the commissioning phase (Jan – end April 2018) has been made available from the S-5p mission performance centre (MPC) upon acquiring special permission from ESA for this study. The gridded S-5p data (level 3) using the HARP tools are also available from the corresponding author upon request. The CWI data used for creating ground truth has been provided by Meisam Amani. The monthly CWI generated using S-5p data are available in the BIRA-IASB Data Repository

(https://repository.aeronomie.be/) with the identifier https://doi.org/10.18758/71021060.

**Author contributions**

B.G. and M.K.S. designed the research. B.L. and M.K.S. created the level 3 S-5p methane total column and surface albedo data. S.B. ran the classification codes and did the data analysis with inputs from B.G. and M.K.S.. L.G. gave input on wetland emissions and their properties. All authors contributed to the interpretation of the results. M.K.S., B.G., and S.B. wrote the

manuscript with input from all co-authors.

**Competing interests**

The authors declare no competing interests.

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

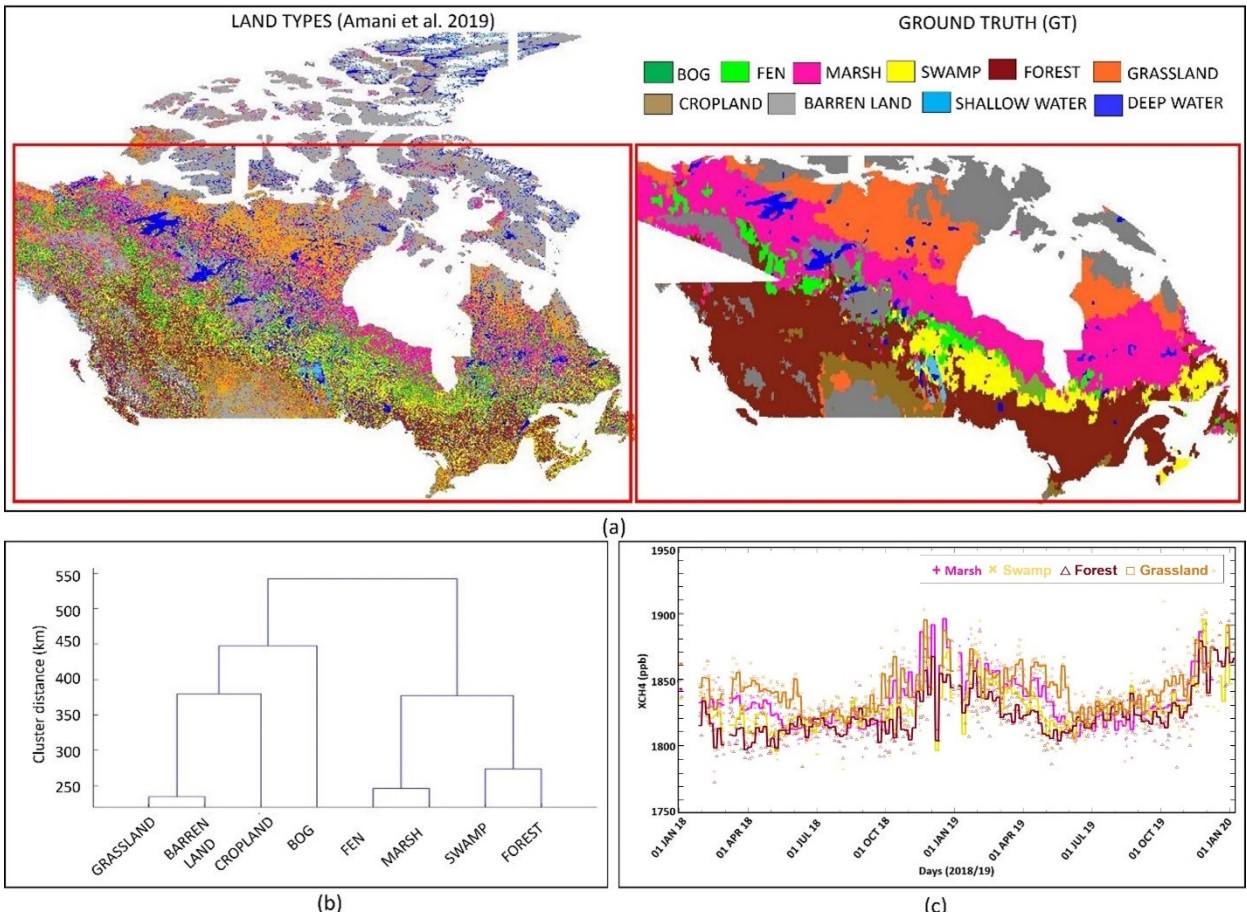

**Figure 1 | Land type classification map (ground truth) creation for Canada and its classification based on S-5p CH₄ product. a, (left)**
**Land types in Canada as described in Amani et al. 2019 (10 classes) at 30 m spatial resolution and (right) ground truth (GT) created**
**using the MODIS NDVI product (Didan (2015)) and graph cut segmentation at 0.05° spatial resolution. The maps were generated**
**using Matlab v.2019b software. The boundary of the map was taken from the open-source website https://www.igismap.com/canada-**
**shapefile-download-free-adminstrative-boundaries-provinces-and-territories/ (last accessed on 15 June 2020). b, An unsupervised**
**clustering of land types, Dendrogram depicting the inbuilt hierarchical relationship that exists in the data (displaying the best**
**division in the land types using Euclidean distance as dissimilarity). c, Time-series (Jan 2018 – Dec 2019) distribution of the CH4**
**(ppb) for the marsh, swamp, forest, and grassland depicting the similarity and points of difference in the dataset. The points are the**
**individual values for the respective days and solid lines are the 5-days running median.**



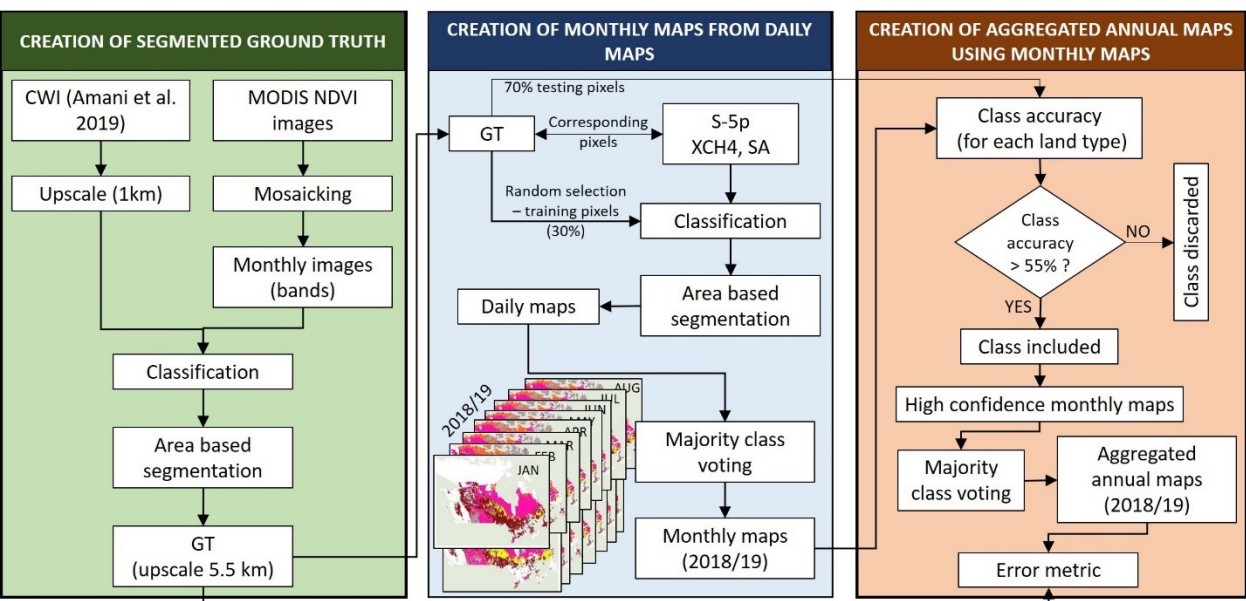

**Figure 2 | Flowchart showing the complete process from the creation of the ground truth (GT) to the creating of annual land type**
**classification maps. The maps were generated using Matlab v.2019b software. The boundary of the map was taken from the open-source website https://www.igismap.com/canada-shapefile-download-free-adminstrative-boundaries-provinces-and-territories/ (last accessed on 15 June 2020).**




**Figure 3 | Time series plots of class accuracies for different land types. Representation of % area covered by snow (dashed red), S-5p (dashed green), accuracy (in %) achieved for land types classification using proposed methodology using S-5p CH₄ (orange), and S-5p CH₄ + SA data (blue) features (left y-axis) and surface albedo (right y-axis) value for each land type over the time period Jan 2018 – Dec 2019 (x-axis). The % area covered by snow is obtained using MODIS daily snow cover product (MOD10A1), the area covered by S-5p is the monthly average of the area captured by S-5p.**



**Figure 4 | Land type maps created using CH4 and SA data as input. a/c, 2018/2019 – seasonal classified maps created using combining the daily images obtained from S-5p, the missing area (white/blank) was not covered by S-5p for that month. b/d, 2018/2019 – aggregated maps created using the pixels with class accuracy ≥ 55% over the months (majority voted) for each year separately. These represent the area with high confidence for both 2018, 2019. The maps were generated using Matlab v.2019b software. The boundary of the map was taken from the open-source website https://www.igismap.com/canada-shapefile-download-free-adminstrative-boundaries-provinces-and-territories/ (last accessed on 15 June 2020).**



**Figure 5 | Time series plots of class accuracies (in %) achieved for different land type classification using proposed methodology using S-5p bias-corrected XCH4 (orange), bias-corrected XCH4 + SA (blue) and surface albedo (SA – green) for each land type over the time period Jan 2018 – Dec 2019 (x-axis).**





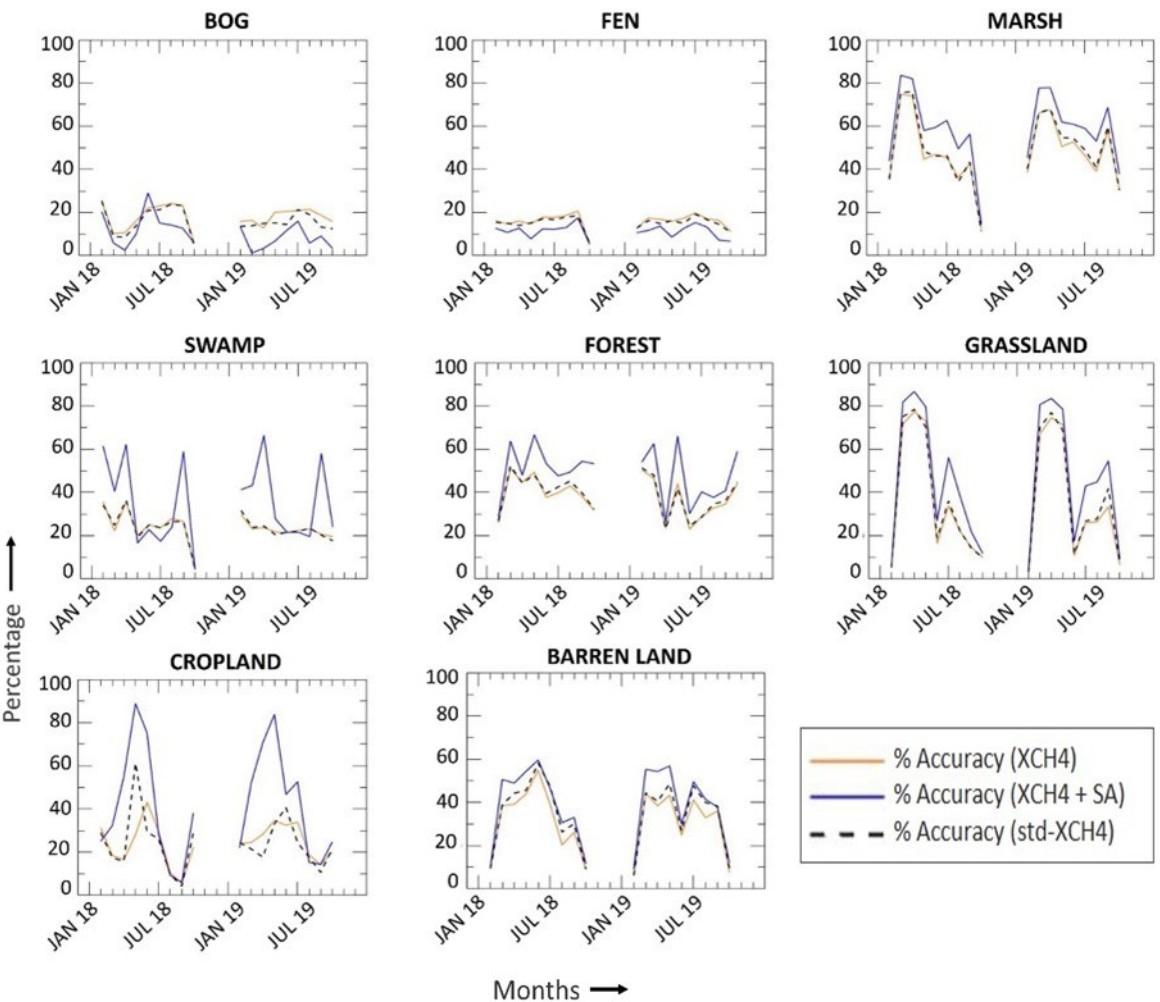


**Figure 6 | Time series plots of class accuracies (in %) achieved for different land types classification using proposed methodology using S-5p bias-corrected XCH4 (orange), bias-corrected XCH4 + SA (blue) and standard XCH4 (std-XCH4 – dashed black) for each land type over the time period Jan 2018 – Dec 2019 (x-axis).**



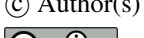

**Table 1 | Geometrical error metric and accuracies for the union of the area covered in aggregated maps of 2018, 2019 (confident land types). The % Δ signifies the absolute change in the parameter value being identified in comparison to the original value of the parameter in the GT. ΔArea signifies the change in the geometrical area. Producer and User accuracy on the GT made using Amani et al. (2019) (2018/19)**

| Aggregated Maps (2018/19) | Identified region | Yr. 2018/19 | Yr. 2018/19 | |
|---|---|---|---|---|
| Land types | Area (1,000 km$^2$) | % ΔArea | Producer Accuracy (%) | User Accuracy (%) |
| MARSH | 602.22 | 13.90/0.89 | 94.01/96.07 | 81.56/97.36 |
| SWAMP | 131.63 | 28.67/30.23 | 65.52/56.41 | 96.63/85.31 |
| FOREST | 544.98 | 9.02/12.23 | 87.02/86.37 | 96.16/100 |
| GRASSLAND | 354.01 | 2.99/9.67 | 90.23/89.17 | 87.44/100 |
| CROPLAND | 75.81 | 20.10/20.21 | 70.85/78.12 | 91.63/100 |
| BARREN LAND | 334.86 | 39.09/16.54 | 47.47/81.47 | 100/100 |


**Table 2 | Confusion Matrix for 2018 and 2019. The x-axis describes the Predicted Class, and the y-axis describes the True Class.**

| | BOG | FEN | MARSH | SWAMP | FOREST | GRASSLAND | CROPLAND | BARREN LAND |
|---|---|---|---|---|---|---|---|---|
| **2018** | | | | | | | | |
| **BOG** | 625 | 4 | 1656 | 158 | 928 | 22 | 9 | 115 |
| **FEN** | 5 | 1907 | 4833 | 458 | 4203 | 59 | 7 | 98 |
| **MARSH** | 18 | 64 | 63601 | 409 | 3700 | 2903 | 44 | 759 |
| **SWAMP** | 11 | 31 | 3707 | 9426 | 7915 | 102 | 28 | 164 |
| **FOREST** | 15 | 45 | 4705 | 446 | 62585 | 314 | 369 | 556 |
| **GRASSLAND** | 2 | 4 | 3840 | 61 | 454 | 30272 | 510 | 2573 |
| **CROPLAND** | 3 | 3 | 3129 | 64 | 780 | 378 | 6263 | 806 |
| **BARREN LAND** | 5 | 11 | 6876 | 165 | 1575 | 3987 | 433 | 33134 |
| **2019** | | | | | | | | |
| **BOG** | 403 | 3 | 2282 | 92 | 917 | 81 | 30 | 21 |



| | | | | | | | |
|---|---|---|---|---|---|---|---|
| **FEN** | 2 | 1820 | 5347 | 176 | 4364 | 136 | 38 | 48 |
| **MARSH** | 10 | 59 | 69014 | 208 | 4266 | 2695 | 99 | 714 |
| **SWAMP** | 10 | 33 | 5033 | 8568 | 7853 | 82 | 70 | 70 |
| **FOREST** | 9 | 38 | 5651 | 267 | 59373 | 437 | 353 | 341 |
| **GRASSLAND** | 1 | 3 | 4266 | 11 | 403 | 31057 | 455 | 2329 |
| **CROPLAND** | 1 | 1 | 2479 | 14 | 999 | 497 | 6628 | 986 |
| **BARREN LAND** | 2 | 10 | 6158 | 47 | 1737 | 4010 | 436 | 34145 |

**Table 3 | Geometrical error metric and accuracies for the union of the area covered in aggregated maps of 2018, 2019 (confident land types). Jaccard represents the 2D similarity when maps are overlapped directly. ΔExtent change in extent, and ΔOrientation change in orientation.**

| Aggregated Maps (2018/19) | Present study | | | | |
|---|---|---|---|---|---|
| | 2018/19 | Identified region | 2018/19 | Identified region | 2018/19 |
| Land types | Jaccard | Orientation (°) | % ΔOrientation | Extent | % ΔExtent |
| MARSH | 0.79/0.94 | -11.61 | 1.89/0.04 | 0.104 | 2.61/1.20 |
| SWAMP | 0.68/0.54 | -4.33 | 9.80/15.40 | 0.059 | 25.59/37.79 |
| FOREST | 0.85/0.88 | -9.9 | 5.91/0.045 | 0.119 | 10.92/11.76 |
| GRASSLAND | 0.82/0.90 | -9.86 | 3.74/0.82 | 0.092 | 1.62/10.27 |
| CROPLAND | 0.70/0.80 | -15.5 | 4.87/8.68 | 0.027 | 10.29/22.86 |
| BARREN LAND | 0.51/0.83 | -7.36 | 34.43/9.67 | 0.071 | 38.16/16.01 |