# Peer review of "Variations in land types detected using methane retrieved from spaceborne sensor"

_Biogeosciences, 2022_

## Author Comment (AC1)

Response to comments from Reviewer 1

Black: Reviewer's comments; Blue: Author's answers; Green: Changes in the manuscript

We thank the reviewer for the review and for providing useful feedback, which we consider in the revised version of the paper. Following is the point-by-point response and account of changes made in the manuscript.

Reviewer's comments:
Bhatnagar et al., used the official operational methane data product from Sentinel-5-Presursor (S-5p) to detect land types in Canada using a machine learning algorithm. Their analysis shows (see Abstract) "unique sensitivity to certain land types". They found (see Abstract) that "the areal extent of six land types (marsh, swamp, forest, grassland, cropland, and barren-land)" can be identified "with high overall accuracy by analysing S-5p data over Canada utilising" their classification-segmentation algorithm. For this purpose, they analysed retrieved methane and retrieved surface albedo individually and in combination. They summarized their results as follow: "Monthly and yearly inventory maps were created, which can be used to validate or complement global models where data from other sources are missing and may help in further constraining the methane budget".

General:

I am very surprised by this study. I don't think that the interpretation w.r.t. methane is correct. It is shown in several recent papers that the operational S-5p methane data product suffers from albedo related methane biases, e.g., Barré et al. (2021), Hachmeister et al., (2022), Lorente et al., (2022) explaining, for example, that the locally elevated methane feature discussed in Froitzheim et al., (2021) is a surface albedo related retrieval artifact. The latest version of the scientific retrieval algorithm of SRON (Lorente et al., 2022) and Univ. Bremen (Schneising et al., 2022) are also addressing this albedo (or spectral surface reflectivity related) issue. Bhatnagar et al. are not citing these papers although they are highly relevant for their work. As surface reflectivity related issues are not mentioned in Bhatnagar et al., I assume that they are not aware of this issue.

As a consequence, it appears that Bhatnagar et al. is misinterpreting the albedo related methane bias as a geophysically interesting methane signal, which can be exploited to get land type information. While it may be true that land type information can be obtained by exploiting the albedo related bias (including possibly also real methane variations related to land type dependent methane emissions), I doubt that their results will helps to "further constraining the methane budget" (as written in their Abstract). I see this study as a detailed and interesting investigation of albedo related biases but not as a study that contributes directly to improving our knowledge on methane sources.

I recommend that the authors carefully study the listed references, cite them and modify the paper accordingly (especially the methane related interpretation and conclusions). I also strongly recommend to analyse in addition the latest versions of the two alternative scientific S-5p XCH4 data products, namely the one from SRON (Lorente et al., 2022) and the one Univ. Bremen (Schneising et al., 2022) to find out to what extent the conclusions are robust w.r.t.

the used data product. I expect that such an analysis would result in significantly different conclusions.

Author's response:
Thank you for the detailed comments and suggestions. We think some excellent points were made, and we have made significant changes in the manuscript accordingly. In the revised version, we have analysed the latest version of the operational S-5p data for four full years (2019, 2020, 2021 and 2022) of available data. We excluded the year 2018, as the RPRO (re-processed version 02.04.00) operational product is available only since 30 April 2018 and not for the full year. Indeed, we believe that a large amount of land variation picked up is contributed by surface albedo. However, in the new results, we also see that XCH4 alone picks up variations, albeit with more modest accuracy; hence, giving some idea of change in methane sensitivities across the land types. We have changed the title and main conclusion of the paper to reflect this.

We have added the references to the literature pointing to the albedo dependence of the S-5p methane product. However, we refrain to use the scientific products of SRON and Univ. Bremen as we find this product comparison is beyond the scope of our paper and should rather be done by the product developers themselves on a global level and not the product users doing the comparison on a regional level as our study area is focusing on Canada only.

Reviewer's comments:
Specific:

Line 47: Unclear for me why a few km resolution atmospheric data product of a long-lived gas can be used to better define the areal extent of different land use types (compared to few 10 m resolution sensors optimized for land applications).

Author's response:
The Introduction section has been rephrased now, removing such confusing statements.

Reviewer's comments:
Line 60 following: The cited reference for the operational algorithm is the pre-launch description and does not reflect the latest version. Please cite also the latest (relevant) ATBD and explicitly mention which version number of the data product has been used.

Author's response:
Done

Reviewer's comments:
Line 65 following: The sparse TCCON network does not permit to validate the accuracy of spatial XCH4 maps and, therefore, the listed results in terms of systematic uncertainty may be too optimistic for the application addressed in this publication. I recommend to add this caveat.

Author's response:
Added a sentence in section 2 to reflect this point.

"However, the network is relatively sparse with gaps in many regions of the world and do not cover the complete range of surface conditions. So the uncertainty values stated corresponds to only the characteristics of the sites where the TCCON stations are located and similar situations elsewhere."

Reviewer's comments:
Line 93: Please explain "producer accuracy" and "user accuracy".

Author's response:
Definition added in section 2.2.2

"The producer accuracy is accuracy that relates to correct classification of a class by the algorithm and the user accuracy depicts the reality on the ground."

Reviewer's comments:
Equation (2): Please explain all abbreviations (TP, FN, …).

Author's response:
Description added in section 3.2.

"Class Accuracy (CA) is the ratio of the diagonal vector of the class under consideration (true positives (TP) with the total number of pixels belonging to the same class (true positives (TP) and false negatives (FN)"

Reviewer's comments:
Line 154: Please explain "kappa value".

Author's response:
Based on Foody (2020), the usage of the kappa value has now been dropped, hence, it is not described further.

Foody, G. M. (2020). Explaining the unsuitability of the kappa coefficient in the assessment and comparison of the accuracy of thematic maps obtained by image classification. *Remote sensing of environment*, *239*, 111630.

Reviewer's comments:
Section 3.3: Please provide a more detailed explanation of the error metric (J, A, O, E) including how the results are to be interpreted when presenting Table 3.

Author's response:
Description added in section 3.3.

"The spatial error metrics treat each class as a segment, and accuracy is determined using the intersection of the segments with the ground truth. The segment-based approach improves the interpretation of the data, as opposed to the conventional pixel-based confusion matrix, by computing error diagnostics based on the distance of the segments. In reality, the land type can be changed/shifted/grown. This change is penalised heavily in a pixel-based approach,

whereas, in a segment-based approach, such changes are weighted and can be forfeited based on the amount of change and interpretation of it."

Reviewer's comments:
Figure 2: Very nice and informative !

Author's response:
Many thanks.

Reviewer's comments:
Figures 3, 5, 6: Please explain better the various curves shown in Figure 3 (how have they been computed, what do they show, interpretation for the purpose of the presented study; I recommend to use one or two cases (e.g., BOG and GRASSLAND) to explain as clearly as possible).

Author's response:
Description added in section 4.

"The time-series of the class accuracy (CA) values for each land type calculated compared to the GT maps are shown in Fig. 3, along with the SA for the respective land types. The figure also shows how the accuracy is affected by S5-p coverage and changes drastically even when the SA is almost constant throughout the years. The % accuracy depicts the CA (Eq. 2), and the SA (grey line) is the spatial mean for each class each day. Taking the example of Bog land type, we can see the SA remains approximately constant over the year (which is expected). However, there is varying accuracy when using SA as a variable in the classification algorithm (blue line). This therefore, shows some independent effect of methane on the land-type variability. The cropland class, in contrast, has a more variable SA through the years – this is also due to the presence of higher levels of snow in the area and eventual residuals. The addition of SA, in this case, definitely complements the land-type analysis. However, looking at the variation in the methane-only accuracy of the algorithm (orange line), such high dips also concur with our previously stated speculation. The performance metrics are further discussed in detail in the following paragraphs."

Reviewer's comments:
Typos etc.:

Line 50: Replace S5 by S-5p.

Author's response:
Corrected.

---

## Author Comment (AC2)

**Response to comments from Editor (Dr. Jamie Shutler)**

Black: Reviewer's comments; Blue: Author's answers; Green: Changes in the manuscript

**Dear Editor,**

We thank you for handling the manuscript and providing careful guidance to us.

We accept your comments and have included the suggested changes in the manuscript. Following is the point-by-point response and account of changes made in the manuscript.

**Editor's comments:**

I read your paper with interest as it is excellent to see satellite column integrated gas observations being used within a biogeoscience study. I have had significant issues in identifying reviewers for your manuscript, having invited 21 reviewers, 4 of which accepted, but then only 1 reviewer submitted a report. Hence I am now submitting this editor comment so that we can allow this review process to proceed. I realise that I have previously reviewed your paper prior to its publication within the discussion forum and that you revised your work addressing my earlier comments. So my comments below focus mainly on the major points raised by the single reviewer.

Its clear from the reviewer's comment that your manuscript has suffered from some unfortunate timing in relation to your analysis and then the subsequent release of an updated Sentinel 5P methane dataset. The production of this revised Sentinel 5P methane dataset was triggered by an error (regional bias) that was identified within these data (as presented most recently within Lorente et al., 2022, but also studied within the three other references identified by the reviewer). And it appears that this bias likely forms part of the signal identified within your analysis and manuscript. And you have (not surprisingly) attributed the signal to a change in the natural system, whereas it seems highly likely that at least a part of the signal you identify is due to the error within the Sentinel 5P methane data dataset. The updates and changes in this underlying Sentinel 5P dataset are likely to significantly impact your results and therefore the conclusions from your work are also likely to change.

In light of this, its clear that you should at least repeat your analysis using the updated datasets (i.e. those provided by the reviewer) and then revise your manuscript following the results of this new analysis. I therefore conclude that major revisions are required.

You can re-submit your analysis that use the most recent datasets, revise your conclusions and you may have to revise your paper title. If you choose to perform these major revisions you will need to make sure that you fully account for the new revised Sentinel 5P data along with the associated data uncertainties and make sure that you show how these uncertainties likely impact your results. This will help to illustrate how robust your findings are to the underlying uncertainties of the Sentinel 5P dataset. This issue of unfortunate timing highlights the need to

include the data version numbers and sources for all data (so authors can trace which datasets were used) so please make sure you include this information within your revised work.

**Author's comments:**

We are very thankful for your comments, suggestions, and consideration of our manuscript. We had used all available S-5p methane operational data at that time in the initial version of the paper. This consisted of data versions 01.02.02 up to 01.03.02 for both reprocessed and offline data (see product readme file for details on the processor version and changes https://sentinels.copernicus.eu/documents/247904/3541451/Sentinel-5P-Methane-Product-Readme-File.pdf/d7214038-25a9-416f-8deb-d5d6c766f92e?t=1678985481272). The current versions (02.04.00 and 02.05.00) have undergone significant improvements in methane product quality based on changes in the spectroscopy, bias-correction method, surface reflectance spectral dependence using a higher order polynomial fit and so on. These changes were described in the scientific version of the SRON product (Lorente et al. 2022) and the latest version of the ATBD. Based on the validation results using TCCON data as reference the residual systematic and random uncertainty of the latest S-5p methane data are well within the mission requirement. However, as also pointed by the reviewer 1, these stations are sparse and there remains holes in critical regions of the world. Further extension of the ground-based network would help to better evaluate and constrain the biases of the satellite products in the future. We have used the new available operational S-5p data for all 4 years (2019-22) to show the robustness of our methodology. We have updated the title and the conclusions of the paper according to our new findings.

We have attached the modified version of the manuscript for your evaluation (in track changes mode so that changes become apparent).

We hope that the changes made are to the satisfaction of the reviewers. Thank you in advance for considering the manuscript, and please, contact us if you have any questions or need any further information.

Yours Sincerely, Saheba Bhatnagar and Mahesh Kumar Sha (on behalf of all authors)